# Disentangling Representations of Text by Masking Transformers

## Abstract

Representations from large pretrained models such as BERT encode a range of features in a single vector that affords strong predictive accuracy on a multitude of downstream tasks. In this paper we explore whether it is possible to learn *disentangled representations* by identifying existing subnetworks within pretrained models that encode distinct, complementary aspect representations. Concretely, we learn binary masks over transformer weights or hidden units to uncover the subset of features that correlate with a specific factor of variation; this eliminates the need to train a disentangled model from scratch for a particular domain. We evaluate the ability of this method to disentangle representations of syntax and semantics, and sentiment from genre in the context of movie reviews. By combining masking with magnitude pruning we find that we can identify sparse subnetworks within BERT that strongly encode particular aspects (e.g., movie sentiment) while only weakly encoding others (movie genre). Moreover, despite *only* learning masks, we find that disentanglement-via-masking performs as well as — and often better than — previously proposed methods based on variational autoencoders and adversarial training.

## 1 Introduction and Motivation

Large-scale pretrained models such as ELMo (Peters et al., 2018), BERT (Devlin et al., 2018), and XLNet (Yang et al., 2019) have come to dominate in modern natural language processing (NLP). Such models rely on self-supervision over large datasets to learn general-purpose representations of text that achieve competitive predictive performance across a spectrum of downstream tasks (Liu et al., 2019). A downside of such learned representations is that it is not obvious what information they are encoding, which hinders model robustness and interpretability. The opacity of representations produced by models such as BERT has motivated a line of NLP research on designing probing tasks as a means of uncovering what properties of input texts are encoded into token- and sentence-level representations (Rogers et al., 2020; Linzen et al., 2019; Tenney et al., 2019).

In this paper we investigate whether we can uncover *disentangled representations* from pretrained models. That is, rather than mapping inputs onto a single vector that captures arbitrary combinations of features, our aim is to extract a representation that factorizes into distinct, complementary properties of the input. An advantage of explicitly factorizing representations is that it aids interpretability, in the sense that it becomes more straightforward to determine what factors of variation inform predictions in downstream tasks. Further, disentangled representations may facilitate increased robustness under distributional shifts by capturing a notion of invariance: If syntactic changes in a sentence do not affect the representation of semantic features, and vice versa, then we can hope to learn models that are less sensitive to any incidental correlations between these factors. A general motivation for learning disentangled representations is to try and minimize — or at least expose — model reliance on spurious correlations, i.e., relationships between (potentially sensitive) attributes and labels that exist in the training data but which are not casually linked (Kaushik et al., 2020). This is particularly relevant in the context of large pretrained models like BERT, as we do not know what the representations produced by such models encode.

To date, most research on disentangled representations has focused on applications in computer vision (Locatello et al., 2019b; Kulkarni et al., 2015; Chen et al., 2016; Higgins et al., 2016), where there exist comparatively clear independent factors of variation such as size, position, color, and

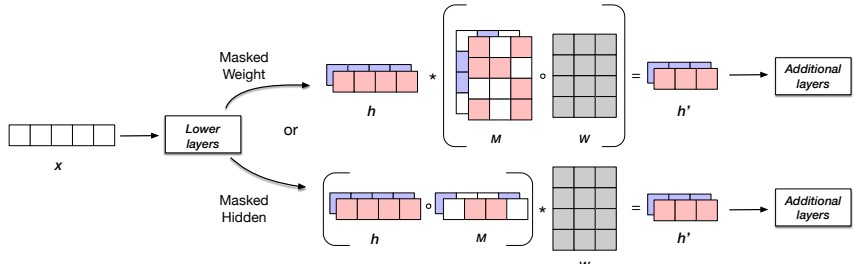

Figure 1: Masking weights and hidden activations in BERT. We show a linear layer with weights $W$, inputs $h$, and outputs $h'$. We learn a mask for each disentangled factor, which is either applied to the weights $W$ or to intermediate representations $h$.

orientation, which have physical grounding and can be formalized in terms of actions of symmetry subgroups (Higgins et al., 2018). A challenge in learning disentangled representations of text is that it is more ambiguous what factors of variation should admit invariance. Still, we may hope to disentangle particular properties for certain applications (e.g., sentiment, or perhaps protected demographic information (Elazar & Goldberg, 2018)), and there are also general properties of language that we might hope to disentangle, e.g., syntax and semantics (Chen et al., 2019).

In this paper we ask whether complementary factors of variation might already be captured by pretrained models, and whether it is possible to uncover these by identifying appropriate "subnetworks". The intuition for this hypothesis is that generalization across a sufficiently large and diverse training set may implicitly necessitate representations that admit some notion of invariance, as the many factors of variation in the training data give rise to a combinatorial explosion of possible inputs. Intriguing prior work (Radford et al., 2017) examining the correlation between sentiment and individual nodes within pretrained networks offers some additional support for this intuition.

To test this hypothesis, we propose to use masking as a mechanism to isolate representations of individual factors. Recent work on *lottery tickets* (Frankle & Carbin, 2018) suggest that overparameterized networks are redundant, in that a network reduced to a small subset of weights set to "winning" initial values can achieve predictive performance similar to the full network. Building on this intuition, we hypothesize that it might be possible to uncover a representation for a factor of interest by starting with a pretrained representation and simply masking out weights or hidden units that correlate with other factors of variation.

We use BERT (Devlin et al., 2018) as an archetypal pretrained transformer to test two variants of this basic idea. In the first variant we learn binary masks for all *weight matrices* in the model; in the second we derive masks for all *hidden units* (intermediate representations). To learn these masks we minimize a triplet loss that encourages the resultant representations for instances that are similar with respect to an aspect of interest to be relatively near to one another, independent of other factors. Our approach of uncovering existing subnetworks within pretrained models that yield disentangled representations differs substantially from prior work on disentangling representations in NLP, which have either relied on adversarial debiasing approaches (Elazar & Goldberg, 2018; Barrett et al., 2019) or variational auto-encoders (Chen et al., 2019; Esmaeili et al., 2019).

We evaluate masking in the context of two tasks. The first is a setting in which we aim to disentangle a representation of features for a target task from that of information encoding a secondary, non-target attribute (e.g., this might be sensitive information, or simply an unrelated factor). In the second we follow prior work in attempting to induce representations of semantics and syntax, respectively. In both settings, our surprising finding is that masking alone often outperforms previously proposed approaches (which learn or finetune networks in their entirety). While a small amount of masking generally suffices to achieve disentanglement, we can further increase sparsity by combining masking with weight pruning.

The main contributions of this paper are as follows. (1) We propose a novel method of disentangling representations in NLP: Masking weights or hidden units within pretrained transformers (here, BERT). (2) We empirically demonstrate that we are indeed able to identify sub-networks within pretrained transformers that yield disentangled representations that outperform existing approaches

(which finetune all model weights). (3) We show that masking can be combined with weight-pruning techniques to learn sparse subnetworks for each factor of interest.

## 2    METHODS

We are interested in learning a disentangled representation that maps inputs $x$ (text) onto vectors $z^{(a)}$ and $z^{(b)}$ that encode two distinct factors of variation. To do so, we will learn two sets of masks $M^{(a)}$ and $M^{(b)}$ that can be applied to either the weights or the intermediate representations in a pretrained model (in our case, BERT). We estimate only the mask parameters and do not finetune the weights of the pretrained model.

To learn the masks $M^{(a)}$ and $M^{(b)}$, we assume access to triplets $(x_0, x_1, x_2)$ in which $x_0$ and $x_1$ are similar with respect to aspect $a$ but dissimilar with respect to aspect $b$, whereas $x_0$ and $x_2$ are similar with respect to aspect $b$ but dissimilar with respect to aspect $a$. In some of our experiments (i.e., when disentangling the sentiment from the genre in movie reviews) we further assume that we have access to class labels $y^{(a)} \in \{0, 1\}$ and $y^{(b)} \in \{0, 1\}$ for the aspects of interest.

### 2.1    MASKING WEIGHTS AND HIDDEN ACTIVATIONS

Figure 1 illustrates the two forms of masking that we consider in our approach. We depict a single linear layer of the model, $h = (h^{(a)}, h^{(b)})$ are the input activations, $W$ are the weights in the pretrained model,[1] and $h' = (h'^{(a)}, h'^{(b)})$ are the output activations. We augment each layer of the original network with two masks $M = (M^{(a)}, M^{(b)})$, applied in two ways:

**1. Masking Weights**    When training BERT with masked weights, masks $M^{(a)}$ and $M^{(b)}$ have the same shape as weights $W$, and output activations are computed using the masked tensor of weights

$$h' = h \cdot (W \circ M). \tag{1}$$

**2. Masking Hidden Activations**    When training BERT with masked hidden units, the masks $M^{(a)}$ and $M^{(b)}$ have the same shape as the intermediate (hidden) activations $h^{(a)}$ and $h^{(b)}$. Output activations are computed by applying the original weights $W$ to masked inputs

$$h' = (h \circ M) \cdot W. \tag{2}$$

In both methods, we follow Zhao et al. (2020) and only mask the last several layers of BERT, leaving the bottom layers (including the embedding layer) unchanged. In practice we find masking the last 9 layers to work the best.

### 2.2    TRIPLET LOSS

To learn the masks, we assume that we have access to supervision in the form of triplets, as introduced above. Passing $(x_0, x_1, x_2)$ through our model yields two representations of each instance: $(z_0^{(a)}, z_0^{(b)}), (z_1^{(a)}, z_1^{(b)}), (z_2^{(a)}, z_2^{(b)})$, for which we define the loss

$$L_{\text{triplet}}^{(a)} = \max\left(\|z_0^{(a)} - z_1^{(a)}\| - \|z_0^{(a)} - z_2^{(a)}\| + \epsilon, 0\right), \tag{3}$$

$$L_{\text{triplet}}^{(b)} = \max\left(\|z_0^{(b)} - z_2^{(b)}\| - \|z_0^{(b)} - z_1^{(b)}\| + \epsilon, 0\right), \tag{4}$$

$$L_{\text{triplet}} = \frac{1}{2}\left(L_{\text{triplet}}^{(a)} + L_{\text{triplet}}^{(b)}\right). \tag{5}$$

Here $\epsilon$ is a hyperparameter specifying a margin for the loss, which we set to $\epsilon = 2$ in all experiments.

### 2.3    SUPERVISED LOSS

In some settings we may have access to more direct forms of supervision. For example, when learning representations for the genre and sentiment in a movie review, we have access to class

---

[1]We omit the bias term in the linear layer, for which no masking is performed.

labels $y^{(a)}$ and $y^{(b)}$ for each aspect. To make full use of such supervision when available, we add two linear classification layers $C^{(a)}$ and $C^{(b)}$ to our model and compute the classification losses

$$L_{\text{cls}}^{(a)} = \text{CrossEntropy}\Big(C^{(a)}(z^{(a)}), y^{(a)}\Big), \quad L_{\text{cls}}^{(b)} = \text{CrossEntropy}\Big(C^{(b)}(z^{(b)}), y^{(b)}\Big), \quad (6)$$

$$L_{\text{cls}} = \frac{1}{2}\Big(L_{\text{cls}}^{(a)} + L_{\text{cls}}^{(b)}\Big). \quad (7)$$

### 2.4 DISENTANGLEMET LOSS

To ensure that each of the two aspect representations are distinct, we encourage the masks to be mutually exclusive. That is, masks of different layers should overlap as little as possible. We include an additional term in the loss for each layer $L$ to achieve this

$$L_{\text{overlap}} = \frac{1}{L}\sum_{l=1}^{L}\sum_{i,j}\mathbb{1}_{(M_{i,j}^{(a)}+M_{i,j}^{(b)}>1)}. \quad (8)$$

### 2.5 BINARIZATION AND GRADIENT ESTIMATION

The final loss of our model is a weighted sum of all the losses

$$L = \lambda_{\text{triplet}} \cdot L_{\text{triplet}} + \lambda_{\text{overlap}} \cdot L_{\text{overlap}}(+\lambda_{\text{cls}} \cdot L_{\text{cls}}). \quad (9)$$

Where we parenthetically denote the cls loss, which we will only sometimes have. We minimize this loss with respect to $M$ (and the classifier parameters), whilst keeping pretrained BERT weights fixed. Because the loss is not differentiable with respect to a binary mask, we learn continuous masks $M$ that are binarized during the forward pass by applying a threshold $\tau$, which is a global hyperparameter,

$$M_{ij}^* = \begin{cases} 1 & \text{if } M_{ij} \geq \tau \\ 0 & \text{if } M_{ij} < \tau \end{cases} \quad (10)$$

We then use a *straight-through* estimator (Hinton et al., 2012; Bengio et al., 2013) to approximate the derivative, which is to say that we evaluate the derivative of the loss with respect to the continuous mask $M$ at the binarized values $M = M^*$,

$$M = M - \eta \left.\frac{\partial L}{\partial M}\right|_{M=M^*}. \quad (11)$$

## 3 EXPERIMENTS

### 3.1 DISENTANGLING SENTIMENT FROM GENRE IN MOVIE REVIEWS

**Experimental Setup** In this experiment we assume a setting in which each data point $x$ has both a 'main' label $y$ and a secondary (possibly sensitive) attribute $z$. We are interested in evaluating the degree to which explicitly disentangling representations corresponding to these may afford robustness to shifts in the conditional distribution of $y$ given $z$. As a convenient, illustrative dataset with which to investigate this, we use a set of movie reviews from IMDB (Maas et al., 2011) in which each review has both a binary sentiment label and a genre label.

Here we consider just two genres: Drama and Horror. We exclude reviews corresponding to other genres, as well as the (small) set of instances that belongs to both genres. To investigate robustness to shifts in correlations between $z$ and $y$ we sampled two subsets of the training set such that in the first sentiment and genre are highly correlated, while in the second they are uncorrelated. We report the correlations between these variables in the two subsets in Table 1. We train models on the correlated subset, and then evaluate them on the uncorrelated set.

We compare the proposed masking approaches to several baselines. **Untuned** corresponds to a dense classification layer on top of representations from BERT (without finetuning). In the **finetuned** variant we omit masks and instead minimize the loss with respect to BERT weights. In the **adversarial** model we adopt adversarial 'debiasing': In addition to minimizing loss on the main task, here we also train an adversarial classifier to predict the secondary (non-targeted) attribute, and the encoder is trained to mitigate the adversaries ability to do so. We implement this via the gradient-reversal method described by Ganin & Lempitsky (2015). We also compare with two variational autoencoder baselines, **VAE** is a VAE model with multi-task loss and adversarial loss, introduced in John et al. (2019), and **VAE+BERT** is the same model but using BERT as the embedding module.

Table 1: Conditional prevalences of the respective classes in the correlated (left) and uncorrelated (right) datasets. We control for the total number of positive and negative samples to be equal in both datasets, so both datasets are balanced with regard to sentiment and with regard to genre. We use the correlated dataset for training, but evaluate on the uncorrelated dataset, to ensure that the prediction of the model reflects the aspect of interest, rather than correlations between aspects.

| **Training: Correlated** | | | | **Test: Uncorrelated** | | |
|---|---|---|---|---|---|---|
| Sentiment / Genre | Drama | Horror | | Sentiment / Genre | Drama | Horror |
| Positive | 0.85 | 0.15 | | Positive | 0.5 | 0.5 |
| Negative | 0.15 | 0.85 | | Negative | 0.5 | 0.5 |

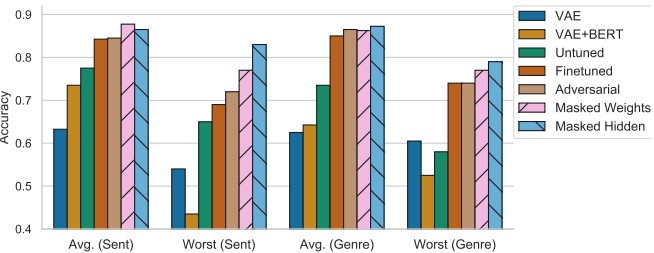

Figure 2: Average and worst main task performance across sentiment/genre combinations. Masked variants (proposed in this paper) are cross-hatched. Larger gaps between the average performance and the worst group performance indicate that the corresponding model is using the non-targeted attribute when making predictions for the main task.

**Leakage of the non-targeted attribute.** We evaluate the degree to which representations "leak" non-targeted information. Following Elazar & Goldberg (2018), we first train the model to predict the main task label on the correlated dataset. Then we fix the encoder and train an *attacker* (a single layer MLP) on the uncorrelated dataset to probe the learned representations for the non-targeted attribute. Because the attacker is both trained and tested on the uncorrelated dataset, it cannot simply learn the main task and exploit the correlation. We show the performance of our proposed masking models and baselines in Table 2. Masking variants perform comparably to baselines with respect to predicting the main task label, but do so with notably less leakage than these methods.

**Performance on worst groups.** In addition to leakage of the non-targeted attribute, we are interested in how models perform on the main tasks for each subgroup: (Positive, Drama), (Positive, Horror), (Negative, Drama), and (Negative, Horror). Because the distribution of the four groups is not uniform in the training set, we expect that models will perform better on combinations that are over-represented in this set, and worse on under-represented attribute combinations. This decline would indicate that a model is implicitly exploiting the correlation between these attributes.

Table 2: Performance on sentiment and genre prediction. We report accuracy on the main task ($\uparrow$ *higher is better*) as well as leakage to the non-targeted task ($\downarrow$ *lower is better*).

| | **Factor: Sentiment** | | **Factor: Genre** | |
|---|---|---|---|---|
| | Main task $\uparrow$ | Leakage $\downarrow$ | Main task $\uparrow$ | Leakage $\downarrow$ |
| VAE | 62.1 | 59.0 | 65.6 | 61.13 |
| VAE + BERT | 67.5 | 66.3 | 71.4 | 70.3 |
| Untuned | 82.3 | 81.5 | 81.5 | 82.3 |
| Finetuned | 87.5 | 85.5 | **87.3** | 86.0 |
| Adversarial | 86.8 | 80.3 | 85.0 | 75.5 |
| Masked Weights | **88.0** | **72.0** | 87.0 | **73.0** |
| Masked Hidden | **88.0** | 79.0 | 85.0 | 79.0 |

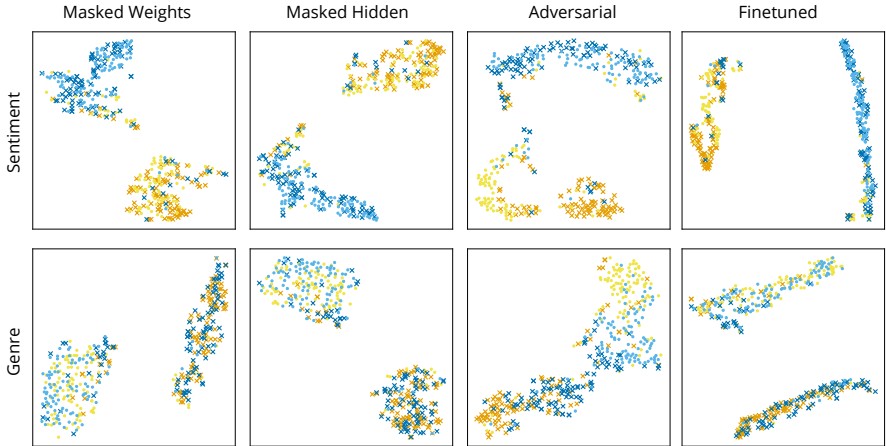

Figure 3: t-SNE projection of sentiment representations and genre representations of different models. Marker colors denote sentiment (**blue** for positive and **yellow** for negative); marker shapes denote genre (× for drama and • for horror).

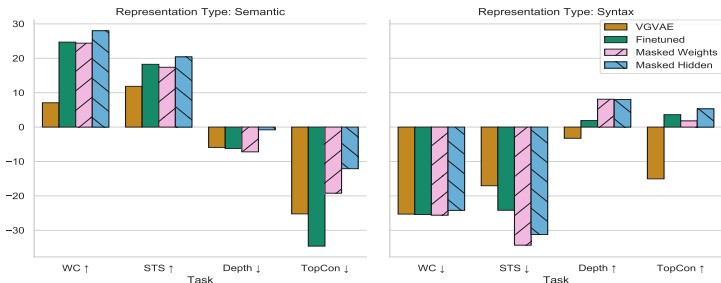

Figure 4: Differences between the performances achieved via BERT embeddings and the disentangled model variants considered on semantics-oriented (WC, STS) and syntax-oriented (Depth, TopCon) tasks compared with BERT embeddings. We plot this difference with respect to the semantics and syntax embeddings induced by the models in the left and right subplots, respectively.

We report both the average performance on the four groups, and the worst performance observed, which is a proxy for lower bound for a model when applied to a dataset where group composition differs from the training dataset. Figure 2 plots the results. Again we observe that the masking variants realize similar average performance as the baselines, but consistently outperform these other models in terms of worst performance. This indicates that the proposed variants rely less on the correlation between the two attributes when predicting the main label, as we would hope.

**Qualitative Evaluation.** In Figure 3 we plot t-SNE visualizations (Maaten & Hinton, 2008) of the representations induced by different models. If the representations are indeed disentangled as desired, instances with different sentiment will be well separated, while those belonging to different genres within each sentiment will **not** be separated. Similarly, for genre representations, we hope to see that instances of the same genre co-locate, but that there is no clustering of examples that reflects sentiment. While no method perfectly realizes these criteria, we observe that the masking approaches we have proposed achieve better results here than do the two baselines. In both Adversarial (Sentiment) and Finetune (Sentiment), instances that have negative sentiment but different genres are still separated, indicating that these sentiment representations still carry genre information.

## 3.2 DISENTANGLING SEMANTICS FROM SYNTAX

**Experimental Setup.** For the second experiment, we follow prior work in attempting to disentangle semantic from syntactic information encoded in learned (BERT) representations of text. Because we have proposed exploiting triplet-loss, we first construct triplets $(x_0, x_1, x_2)$ such that $x_0$ and $x_1$ are similar semantically but differ in syntax, while $x_0$ and $x_2$ are syntactically similar but encode different semantic information. We follow the methods described in prior work on this task (Chen et al., 2019; Ravfogel et al., 2019) to derive triplets. Specifically, we obtain $x_0, x_1$ from the by ParaNMT-

50M (Wieting & Gimpel, 2018) dataset. Here $x_1$ is obtained by applying back-translation to $x_0$, i.e., by translating $x_0$ from English to a different language and then back into English. To derive $x_2$ we keep all function words in $x_0$, and replace all content words by masking each in turn, running the resultant input forward through BERT, and selecting one of the top predictions (that differs from the original word) as replacement tokens.

We compare our disentanglement-via-masking strategies against models that represent state-of-the-art approaches to disentangling syntax and semantics. In particular, we compare against VGVAE (Chen et al., 2019), trained on top of BERT-base to ensure fair comparison. Following prior work that has used triplet loss for disentanglement, we also compare against a model in which we finetune BERT using the same triplet loss that we use to train our model, but in which we update all model parameters (as opposed to only estimating mask parameters).

To evaluate learned representations with respect to the semantic and syntactic information that they encode, we evaluate them on four tasks. Two of these depend predominantly on semantic information, while the other two depend more heavily on syntax.[2] For the semantics tasks we use: (i) A word content (WC) (Conneau et al., 2018) task in which we probe sentence representations to assess whether the corresponding sentence contains a particular word; and (ii) A semantic textual similarity (STS) benchmark (Rosenthal et al., 2017), which includes human provided similarity scores between pairs of sentences. The former we evaluate in terms of accuracy; for the latter (a ranking task) we use Spearman correlation. To evaluate whether representations encode syntax, we use: (i) A task in which the aim is to predict the longest path in a sentence's parse tree from its embedding (Depth) (Conneau et al., 2018); and (ii) A task in which we probe sentence representations for the type of their top constituents immediately below the $S$ node (TopConst).[3]

In Figure 4 we report the signed difference between the performance achieved by BERT embeddings (we mean-pool over token embeddings) and the two representation types induced by the respective (disentangled) models considered. Ideally, the semantics sentence embeddings would do well on the semantic tasks (WC and STS) and poorly on the syntax-oriented tasks (Depth and TopCon); likewise, syntax embeddings should do well on Depth and TopCon, but poorly on WC and STS. We observe that the proposed masking strategy achieves performance at least equivalent to — and sometimes (as in the case of the syntax embeddings), superior to — alternative approaches. We emphasize that this is achieved *only via masking, and without modifying the underlying model weights*.

### 3.3 INCREASING THE SPARSITY OF THE SUB-NETWORKS

Here we set out to determine if we are able to identify *sparse* subnetworks by combining the proposed masking approaches with *magnitude pruning* (Han et al., 2015a). Specifically, we use the loss function defined in Equation 9 to finetune BERT for $k$ iterations, and prune weights associated with the $m$ smallest magnitudes after training. We then initialize masks to the sparse sub-networks identified in this way, and continue refining these masks via the training procedure proposed above. We compare the resultant sparse network to networks similarly pruned (but not masked). Specifically, for the latter we consider: Standard magnitude tuning applied to BERT, without additional tuning (Pruned + Untuned), and a method in which after magnitude pruning we resume finetuning of the subnetwork until convergence, using the aforementioned loss function (Pruned + Finetuned).

We compare the performance achieved on the semantic and syntactic tasks by the subnetworks identified using the above strategies at varying levels of sparsity, namely after pruning: {0, 20%, 40%, 60%, 80%, 85%, 90%, 95%} of weights.[4] We report results in Figure 5. We observe that proposed masking strategy (in concert with magnitude pruning) consistently yields representations of semantics (top row) that perform comparatively strongly on the semantics-oriented tasks (STS, WC), even at very high levels of sparsity; these semantics representations also perform comparatively poorly on the syntax-oriented tasks (Depth, TopCon), as we might hope. Similarly, syntax representations (bottom) perform poorly on the semantics-oriented tasks (and seem to leak less of this information than other methods), and outperform alternatives on the syntax-oriented tasks.

---

[2]We acknowledge that this is a very simplified view of 'semantics' and 'syntax'.

[3]See Conneau et al. (2018) for more details regarding WC, Depth and TopConst.

[4]Technically, in the Pruned + Masked Weights method, refining the masks may change subnetwork sparsity, but empirically we find this to change the sparsity only slightly ($\sim$1% in all of our experiments).

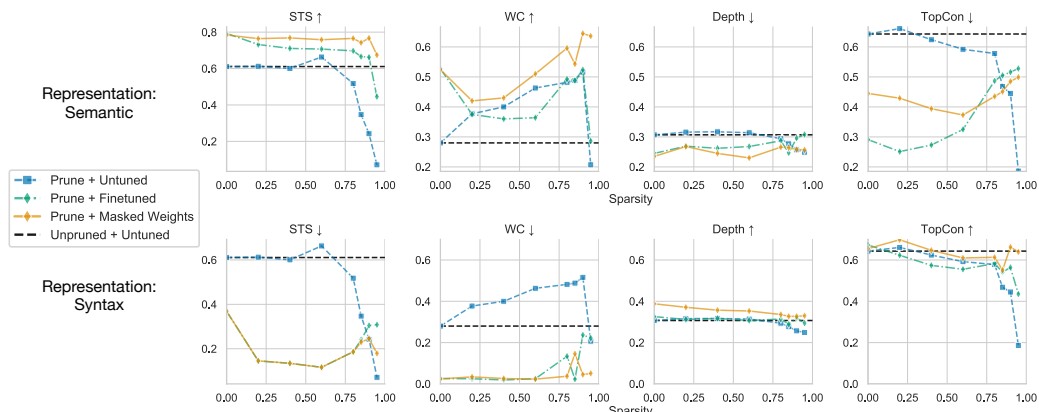

Figure 5: Model performance as a function of the level of pruning. The $x$-axis corresponds to the subnetwork sparsities (percent of weights dropped), while the $y$ axes are performance measures — accuracy for all tasks except for STS, where we report Pearson's correlation. We compare the performance of models trained on the semantic (top) and syntax representations (bottom) learned by the disentangling strategies considered, after pruning to varying levels of sparsity.

## 4 RELATED WORK

**Disentangled and structured representations of images.** The term *Disentangled representations* has been used to refer to a range of methods with differing aims. Much of the initial focus in this space was on learning representations of images, in which certain dimensions correspond to interpretable factors of variation (Kulkarni et al., 2015; Higgins et al., 2016; Chen et al., 2016). In the context of variational autoencoders (Kingma & Welling, 2013; Rezende et al., 2014) this motivated work that evaluates to what extent such representations can recover a set of ground-truth factors of variation when learned without supervision (Eastwood & Williams, 2018; Kim & Mnih, 2018; Chen et al., 2018). Other work has investigated representations with the explicit motivation of fairness (Locatello et al., 2019a; Creager et al., 2019), which disentanglement may help to facilitate.

**Disentangling representations in NLP.** Compared to vision, there has been relatively little work on methods for learning disentangled representations of for natural language data. Much of the prior work on disentanglement for NLP that does exist has focused on using such factored representations to facilitate *controlled generation*, e.g., manipulating sentiment (Larsson et al., 2017). A related notion is that of *style transfer*, for example, separating style from content in language models Shen et al. (2017); Mir et al. (2019). There has also been prior work on learning representations of particular aspects to facilitate domain adaptation (Zhang et al., 2017), and aspect-specific information retrieval (Jain et al., 2018). Esmaeili et al. (2019) focused on disentangling user and item representations for product reviews. Moradshahi et al. (2019) combines BERT with Tensor-Product Representations to improve its transferability across different tasks. Recent work on which we build has proposed learning distinct vectors coding for semantic and syntactic properties of text (Chen et al., 2019; Ravfogel et al., 2019). These serve as baseline models in our experiments.

Finally, while not explicitly framed in terms of disentanglement, efforts to 'de-bias' representations of text are related to our aims. Some of this work has adopted adversarial training to attempt to remove sensitive information (Elazar & Goldberg, 2018; Barrett et al., 2019).

**Network pruning.** A final thread of relevant work concerns selective pruning of neural networks. This has often been done in the interest of model compression Han et al. (2015a;b). Recent intriguing work has considered pruning from a different perspective: Identifying small subnetworks — winning 'lottery tickets' (Frankle & Carbin, 2018) — that, trained in isolation with the right initialization, can match the performance of the original networks from which they were extracted. Very recent work has demonstrated that winning tickets exist within BERT (Chen et al., 2020).

## 5 DISCUSSION

We have presented a novel perspective on learning disentangled representations for natural language processing in which we attempt to uncover existing subnetworks within pretrained transformers

(e.g., BERT) that yield disentangled representations of text. We operationalized this intuition via a *masking* approach, in which we estimate *only* binary masks over weights or hidden states within BERT, leaving all other parameters unchanged. We demonstrated that — somewhat surprisingly — we are able to achieve a level of disentanglement that often exceeds existing approaches (e.g., a varational auto-encoder on top of BERT), which have the benefit of finetuning all model parameters.

Our experiments demonstrate the potential benefits of this approach. In Section 3.1 we showed that disentanglement via masking can yield representations that are comparatively robust to shifts in correlations between (potentially sensitive) attributes and target labels. Aside from increasing robustness, finding sparse subnetworks that induce disentangled representations constitutes a new direction to pursue in service of providing at least one type of model interpretability for NLP. Finally, we note that sparse masking (which does not mutate the underlying transformer parameters) may offer efficiency advantages over alternative approaches.

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
