# OpenReview forum: "Disentangling Representations of Text by Masking Transformers"
_ICLR.cc/2021/Conference — Reject_

### Official Review · AnonReviewer3 · 2020-10-27
**an interesting work on disentangling representations of text**

**Rating:** 5
**Confidence:** 4

**Review:**

The paper proposes a problem of disentangling representations generated in pretraining models, such as BERT. That is, it is possible to learn disentangled representations that encode distinct, complementary aspect representations. To this end, the authors proposes a method that employs the mask technique on transformer weights or hidden units to find the subset of features correlating with a specific task. The experimental results show that the proposed method can encode particular aspects while weakly encoding others. The main contributions of the paper is the introduction of binary masks to identifying some subnetworks, which may correlate with specific tasks, within pretrained models. Overall, the paper is well written and is easy to follow.

Concerns:
1. The experimental setup is not convincing. The authors just consider movie reviews corresponding to Drama and Horror from IMDB and exclude reviews corresponding to other genres. It is obvious that considering only two genres is not convincing and more genres should be considered in the experiments. So, the authors should answer the following questions: (1) Why do the authors just selected these two specific genres to conduct the experiments? (2) Do the authors conduct similar experiments on other genres and what about the experimental results?
2. Figure 3 does not show that proposed method achieves better results than do the two baselines. In fact, the finetuned baseline performs very well according to Figure 3. I suggest that the author adopts some quantitative measures to accurately reflect the differences.
3. In addition, how to use these disentangling representations in downstream tasks, such as text classification, natural language inference, and semantic similarity? It is better to discuss and conduct experiment to show the advantages of their disentangling representations in downstream tasks.

Minor comments:
1. In Formula (9), the parentheses are redundant.

---

> ### Author Response · Authors · 2020-11-13
> **Response to reviewer #3**
>
> We thank the reviewer for their comments and provide clarifications and responses to specific questions below.
>
> **The experimental setup is not convincing. Why only pick the two genres of drama and horror?**
>
> Perhaps we were not clear enough in motivating our experimental setup here: We are interested in examining the degree to which disentanglement via different methods affords robustness to reliance on spurious correlations, such as (in this example) associating a particular genre with a specific sentiment (here, e.g., horror with negative sentiment). In many cases, it is reasonable to assume that either we do not want to rely on certain correlations like this for reasons of fairness, and/or the conditional distributions — p(sentiment|genre) — may shift in the test distribution. With this framing in mind, we selected Drama and Horror because among all the major genres, these two genres of reviews have the most correlation between genre and sentiment: Drama reviews are more likely positive and Horror negative. This is to create a spurious correlation between the genre and sentiment, so that we can probe for robustness to the same.
>
>
>
> **Figure 3 does not show that the proposed method achieves better results than the finetuned baseline**
>
> We feel there is a misreading of the figure here (perhaps we could improve the presentation and description). Figure 3 does show our models outperforming the finetuned baseline, with respect to the representations that it induces. When trained on sentiment (upper row), the representations from the finetuned model are still clustered with respect to genre (marker shape); this clustering is not observed using the proposed masking approaches. When trained on genre (bottom row), the two genres are not well separated; whereas the representations from our two models, although still imperfect, are clearly better separated than the finetuned model. This also aligns with the quantitative results in Figure 2 and Table 2.
>
>
>
> **How to use these disentangling representations in downstream tasks, such as text classification, natural language inference, and semantic similarity? It is better to discuss and conduct experiment to show the advantages of their disentangling representations in downstream tasks.**
>
> We did report the STS-Spearman correlation in Sec 3.2, which is a semantic similarity benchmark. And in Sec 3.1, we designed a specific text classification task with two correlating attributes, which we believe demonstrates the advantage of our model over other baselines with respect to robustness, i.e.,  performing well in situations where other methods result in overreliance on artifacts (spurious correlations) in the training set. We feel these experiments do show the important robustness advantages of this approach in downstream tasks.

---

> > ### Comment · AnonReviewer3 · 2020-11-15
> > **About the response from authors**
> >
> > The response seems reasonable. However, it raises a new question. Since the authors carefully selected the data and  designed specific downstrean tasks, How to ensure or reflect the generality of the proposed method.  That is, what's the meaning of the proposed work.  It is just for specific cases or for general cases.

---

> > > ### Author Response · Authors · 2020-11-15
> > > **Reply to Reviewer#3's response**
> > >
> > > The purpose of the work is to devise a method for disentangling representations of text, and one important potential benefit of such methods is making models more robust, i.e., less reliant on spurious correlations. The method is therefore general in that is appropriate for any application in which robustness is a concern, which in practice is most cases.

---

### Official Review · AnonReviewer1 · 2020-10-29
**Light-weight approach to untangle language model representations**

**Rating:** 6
**Confidence:** 4

**Review:**

This paper proposes a masking strategy to identify subnetworks within language models responsible for predicting different text features. This approach requires no fine-tuning of model parameters and still achieves better results compared to previous approaches. Their experimental results on the movie domain show some level of disentanglement is achieved between sentiment and genre. Disentanglement capabilities of their model between sentence semantics and structure, are also tested on four tasks.

Pros:
- Paper is well-written and the idea is explained well.
- Experiment results are convincing and support the claims.
- Achieving comparable results to SOTA without the need to train or finetune models is interesting especially from a computational point of view.

Cons:
- I wish the authors performed their first experiment on more domains: books, music, etc. and consider more than two labels.
From current results, it's hard to confidently conclude that this approach is generalizable.
- Judging based on Figure 4 results I'm not convinced that the proposed approach does better than the *finetuned* (which I believe has a trained classifier on top of BERT) approach especially for Semantic tasks. Perhaps a discussion/ error analysis would be appropriate given better results on Syntax tasks.
- Also a discussion on the results for masking weights vs. masking hidden units is missing. If I'm not mistaken, mathematically, hidden unit masking is a subset of weight masking, where masking an item in hidden activation is equivalent to masking an entire column in the weight matrix?


Comments:
- Although the idea of masking model parameters to achieve untanglment is new, there has been [previous work](https://www.aclweb.org/anthology/P18-1069.pdf) on using dropout to identify sub-parts of the network that contribute more/ less to model predictions framed as a confidence modeling task. Authors may consider adding it to related work.
- Another missed citation under related work is [HUBERT](https://arxiv.org/pdf/1910.12647.pdf) which examines untanglement of semantics and structure across a wide range of NLP tasks.


Minor typos:
- "we *measure* evaluate them on four tasks ..." on page 7
- "Technically, in the Pruned + Masked Weights method, *the* refining the masks ..."

---

> ### Author Response · Authors · 2020-11-13
> **Response to Review #1**
>
> We thank the reviewer for their detailed comments and suggestions, and respond to all concerns below.
>
> **I wish the authors performed their first experiment on more domains: books, music, etc. and consider more than two labels. From current results, it's hard to confidently conclude that this approach is generalizable.**
>
> We note that we did perform experiments on two different types of datasets, corresponding to (quite) different tasks; we thought this would be more compelling than a suite of experiments on sentiment tasks. We would argue that the fact that our model does well on these two considerably distinct datasets/tasks indicates that it has reasonably good generalizability. We will consider including additional experiments on more datasets for a camera-ready version, however.
>
> **Judging based on Figure 4 results I'm not convinced that the proposed approach does better than the finetuned (which I believe has a trained classifier on top of BERT) approach especially for Semantic tasks. Perhaps a discussion/ error analysis would be appropriate given better results on Syntax tasks.**
>
> The reviewers’  observation is correct: our model performs roughly on par with the fine-tuning method in Figure 4 (arguably a bit better, but the objective is multivariate so it is hard to say). But we would highlight that the main purpose of this paper is to provide a new way of looking at the problem of learning disentangled representations; in Figure 3 we can see that the representations learned using the finetuned approach fail to achieve the level of disentanglement enjoyed by the proposed approach. And again, ours are learned without modifying the BERT weights, which we think is an interesting finding. Furthermore, in the robustness experiments (Figure 2) we show that the fine-tuned approach fares considerably worse than the proposed approach.
>
> **Also a discussion on the results for masking weights vs. masking hidden units is missing. If I'm not mistaken, mathematically, hidden unit masking is a subset of weight masking, where masking an item in hidden activation is equivalent to masking an entire column in the weight matrix?**
>
> In principle, the reviewer is correct: masking hidden units is technically a subset of weight masking. Effectively masking hidden representations is a strategy by which to select grouped sets of weights to mask simultaneously (i.e., all associated with individual nodes), whereas weight masking has more flexibility and no such grouping of masked weights. We therefore think it is intuitive conceptually to think about these as distinct strategies. And from an optimization point of view, masking hidden units may be easier to optimize.
>
> **Reply to comments:**
> We thank the reviewer for the references and will add them to the related work section in the updated version.

---

> > ### Author Response · Authors · 2020-11-19
> > **References added**
> >
> > We have updated our paper and included HUBERT in our related work section.

---

### Official Review · AnonReviewer2 · 2020-10-29

**Rating:** 6
**Confidence:** 4

**Review:**

The paper presents a way to learn disentangled representations with respect to target attributes of interest by learning to mask weights or activations. A particular piece of text is encoded into distinct vectors that capture different factors of variation in the data. The method involves learning masks for each factor of variation while keeping the pre-trained model parameters fixed. The masks for every layer are trained using a combination of a triplet-loss, attribute classification loss, and one that encourages masks for different factors to be different across all layers. The triplet loss forces representations of examples that are similar with respect to a particular attribute to be closer than one that are similar based on another attribute.

Models are evaluated on a sentiment/genre classification on a dataset sampled in such a way that introduces spurious correlations between genre and sentiment but evaluated on data that does not have any such correlation. The approach is also evaluated on disentangling syntax and semantics.

Strengths

Building models that are robust to spurious correlations in data is important for a variety of reasons and learning disentangled representations is a promising way to achieve that. This paper shows good generalization performance on datasets with such characteristics.

The overall approach is simple and only requires training masks over weights/activations at each layer. The masks are trained with a fairly straightforward choice of training objectives.

The paper is well written and the overall approach is easy to understand.

Weaknesses

The triplet loss as formulated in this work seems to make it possible to disentangle only two factors of variation (a) and (b).

There is still a fair amount of attribute leakage and the probe designed to measure this leak is only a single layer MLP, there might be more leakage with stronger probes.

The weight masking strategy significantly increases the number of parameters (although the masks are binary, so it just requires a single bit as opposed to 16/32 bit floating point numbers). In this particular work, the number of parameters triples, and it scales linearly with the number of attributes as well.

It requires running the model forward multiple times to get representations that encode different factors of variation.


Questions & Comments

What would performance look like if masks were trained after fine-tuning on sentiment/genre classification? Rather than training masks directly on top of BERT-base. It would be interesting to see if the model is stable to recover from fine-tuning on data with spurious correlations and still produce disentangled representations.

Is every single weight/activation masked at every transformer layer? The paper seems to lack some specifics about exactly what layers/weights are masked. Along these lines, did you experiment with masking only the last few layers? This could save time & parameters

In Figure 3 is the model training with L_{cls} corresponding to sentiment and then visualized for sentiment and genre? Or is the top trained with the supervised sentiment loss and the bottom for supervised genre loss?

It would be interesting to explore an L1 penalty on the masks for increasing sparsity, possibly in conjunction with magnitude pruning as well.

The WC task doesn't feel very representative of sentence "semantics"

---

> ### Author Response · Authors · 2020-11-13
> **Response to Review #2**
>
> We thank the reviewer for their detailed and insightful comments.  First, we would like to make one clarification on the reviewer's comment:
>
>  **The triplet loss as formulated in this work seems to make it possible to disentangle only two factors of variation (a) and (b).**
>
> It is true we have only shown the approach for two factors, but the method is sufficiently general to be amenable to additional factors, though one would have to construct triplets for all-pairs, which would not scale well to a large number of factors. We view extensions to such cases as an interesting direction for future work.
>
> We respond to all questions and comments below.
>
> **Response to Questions & Comments**
> 1. **What would performance look like if masks were trained after fine-tuning on sentiment/genre classification? Rather than training masks directly on top of BERT-base. It would be interesting to see if the model is stable to recover from fine-tuning on data with spurious correlations and still produce disentangled representations.**
>
> This is an interesting question, and something we did not try. We would conduct more experiments to verify this and update our paper with the results as soon as possible. We are thankful for the suggestion!
>
> 2. **Is every single weight/activation masked at every transformer layer? The paper seems to lack some specifics about exactly what layers/weights are masked. Along these lines, did you experiment with masking only the last few layers? This could save time & parameters**
>
> The reviewer is correct to point out that we should have been more explicit about this; We only mask the last 9 layers of the model (which we found in preliminary experiments on dev data to work well). It appears we omitted this implementation detail and we will clarify this.
>
> 3. **In Figure 3 is the model training with L_{cls} corresponding to sentiment and then visualized for sentiment and genre? Or is the top trained with the supervised sentiment loss and the bottom for supervised genre loss?**
>
> The latter; the top is trained with sentiment loss and the bottom genre loss.
>
> 4.**It would be interesting to explore an L1 penalty on the masks for increasing sparsity, possibly in conjunction with magnitude pruning as well.**
>
> Thank you for this suggestion. We did have an L1-penalty that discourages the mask for different attributes to be on (equal to one) in the same position. The idea there is to encourage mutually exclusive masks. But it would be interesting to also explore an L1 penalty on the masks to improve overall sparsity. We plan to do more experiments on that for the camera-ready version. More generally, we hope this approach suggests a line of alternative sparsity-inducing methods for disentanglement via masking.
>
> 5. **The WC task doesn't feel very representative of sentence "semantics"**
>
> We agree that the degree to which a representation captures “semantics” is hard to measure (or even define). We here follow the setting established in prior work (Conneau et al., 2018) regarding the “semantic” probing tasks. It captures a lexical level of “semantics”, which is often used as a substitute for the real “semantics”. Note that we also use a semantic-similarity task (STS) to better capture the sentence level semantics.

---

### Official Review · AnonReviewer4 · 2020-11-01
**New approach to disentangling representations**

**Rating:** 5
**Confidence:** 3

**Review:**

**Summary**:

The paper proposes a procedure to extract disentangled representations from pretrained BERT models. In particular, the paper proposes learning binary masks over BERT weights (or, as an alternative, over BERT activations) such that the resulting representations correspond to the desired aspect representations. The model requires additional supervision (binary labels or example triplets) and training (for the masks but not the BERT weights). The experiments aim to perform disentangling to ensure that (1) the learned representation does not “leak” a potentially sensitive attribute, and (2) the downstream classifier’s performance is good across all subgroups formed by the attributes. The experiments show that the proposed method outperforms baselines such as unmasked BERT, unmasked-but-finetuned BERT, and unmasked-but-adversarially-finetuned BERT.

**Concerns**:

The fact that one can uncover disentangled representations from BERT models by masking weights/activations is a nice result and I'm not aware of similar approaches for BERT. However, it's unclear from the paper that this approach outperforms previous alternatives:
* First, the abstract mentions that the approach is the same or better than variational auto-encoder approaches, and I don't see it mentioned elsewhere in the main text. Am I missing something?
* Second, the paper does not show improved results on any benchmarks.
As a result, I'm not sure whether the paper will be impactful enough for the community.

**Other Questions**:

* The proposed approach involves training binary masks rather than fine-tuning the BERT weights. Given that the mask has the same shape as the weights, it's unclear whether this is a major speedup. Could you discuss this more?
* Does using the pretrained model (vs. one trained from scratch) help?
* Have you considered masking a subset of the weights/activations (e.g. only in the last layer)?
* Do you have any intuition about the learned masks? E.g. are most weights/activations being removed? How much overlap is there between the masks learned for each attribute?

Overall, I like this research direction, but I think it requires more work to be accepted.

---

> ### Author Response · Authors · 2020-11-13
> **Response to Review #4**
>
> We thank the reviewer for their comments, and we are glad the reviewer found this new direction to be interesting.
> We would like to address two of the main concerns raised in this review, before responding to specific questions:
>
> Lack of comparison to variational auto-encoders: We believe this may be a misunderstanding on the part of the reviewer.  We in fact compare with VGVAE, which is a VAE model, in our experiments involving disentangling semantics and syntax. We show results in Figure 4, which demonstrates that the proposed method outperforms VGVAE.  Because this VGVAE model was specifically designed for disentangling semantics and syntax, we did not include comparisons to it in the sentiment/genre experiment. We agree that a comparison to a (different) VAE-based baseline in the sentiment/genre experiment would strengthen the work, and plan to update the paper with the results from this as soon as possible.
>
> The reviewer correctly points out that we do not achieve “state of the art” (SOTA) on any particular standard benchmark dataset. However this is not the primary aim of this paper. Our interest here is to learn disentangled representations, and there is no existing benchmark for disentanglement in NLP. We aim to achieve this in service of robustness, e.g., to make models less sensitive to spurious correlations (as mentioned by R2). Our experiments are therefore designed to probe the degree to which the proposed approach achieves disentanglement (and robustness); we think (as does R1) that the results are convincing in this respect.
>
> More generally, while achieving SOTA on benchmark datasets is one means of showing the value of particular methods or approaches, we argue that we should not, as a community, require all research to be focussed on topping leaderboards. For example, this would largely preclude any work on robustness and interpretability, which are key open problems in NLP and ML more broadly. (See also Ethayarajh and Jurafsky, 2020: https://arxiv.org/abs/2009.13888).
>
> Response to your other questions:
>
> Q1: Would training binary masks be a speedup over fine-tuning?
> There is no reason to believe that training binary masks will be faster than fine-tuning the model. However, binary masks do have the advantage of requiring less memory, which is often at a premium in GPU-based computations. .
>
> Q2:Does using the pretrained model (vs. one trained from scratch) help?
>
> In our approach we do not fine-tune BERT, so if we randomly initialized this the method would not work. The insight here is to uncover existing subnetworks that yield disentangled representations from pretrained models, so training BERT from scratch would not be a viable approach here.
>
> Q3:Have you considered masking a subset of the weights/activations (e.g. only in the last layer)?
> Great question. We only mask the last 9 layers of the model (which we found in preliminary experiments on dev data to work well). It appears we omitted this implementation detail and we will clarify this.
>
> Q4: Do you have any intuition about the learned masks? E.g. are most weights/activations being removed? How much overlap is there between the masks learned for each attribute?
> For Sec 3.1 and Sec 3.2, only a small percentage of the weights/activations are masked (~1%) at convergence. We note this is an interesting finding in its own right; apparently masking out a small fraction of weights can substantially affect the degree of disentanglement. The overlap is very small between the two attributes (close to 0), which is intuitive.

---

> > ### Author Response · Authors · 2020-11-19
> > **VAE baselines added**
> >
> > We have updated the sentiment/genre experiment (Section 3.1) to include two VAE baselines. As shown in figure 2 and table 2, our methods outperform both baselines.

---

### Decision · Program_Chairs · 2021-01-07
**Final Decision**

**Decision:**

Reject

**Comment:**

This paper explores a methodology for learning disentangled representations using a triplet loss to find subnetworks within a transformer.  The authors compare against several other methods and find that their method performs well without needing to train from scratch. The reviewers thought this paper was well written and the authors were very responsive during the review period.  However, there were some questions about the experimental setup and empirical performance of the paper, leaving the reviewers wondering if the performance was convincing.  We agree that there is value in exploring disentangled representations even if they do not necessarily improve performance (as the authors point out), but clearly explaining the reasoning behind all analyses (e.g. specifically choosing domains to introduce a spurious correlation), and justifying differences in performance is particularly important in these cases.